# Effects and Mechanisms of Exercise on Brain-Derived Neurotrophic Factor (BDNF) Levels and Clinical Outcomes in People with Parkinson’s Disease: A Systematic Review and Meta-Analysis

**DOI:** 10.3390/brainsci14030194

**Published:** 2024-02-21

**Authors:** Daan G. M. Kaagman, Erwin E. H. van Wegen, Natalie Cignetti, Emily Rothermel, Tim Vanbellingen, Mark A. Hirsch

**Affiliations:** 1Faculty of Behavioral and Human Movement Sciences, Vrije Universiteit Amsterdam, 1081 HV Amsterdam, The Netherlands; d.g.m.kaagman@student.vu.nl; 2Department of Rehabilitation Medicine, Amsterdam UMC, Vrije Universiteit Amsterdam, 1081 HV Amsterdam, The Netherlands; 3Amsterdam Movement Sciences, Rehabilitation & Development, Amsterdam UMC, Vrije Universiteit Amsterdam, 1081 HZ Amsterdam, The Netherlands; 4Amsterdam Movement Sciences, Ageing & Vitality, Amsterdam UMC, Vrije Universiteit Amsterdam, 1081 HZ Amsterdam, The Netherlands; 5Amsterdam Neuroscience, Neurovascular Disorders, Amsterdam UMC, University of Amsterdam, 1105 AZ Amsterdam, The Netherlands; 6Department of Physical Medicine and Rehabilitation, Carolinas Medical Center, Atrium Health Carolinas Rehabilitation, Charlotte, NC 28203, USA; natalie.cignetti@atriumhealth.org (N.C.); emily.rothermel@atriumhealth.org (E.R.); mark.hirsch@atriumhealth.org (M.A.H.); 7Gerontechnology and Rehabilitation Group, University of Bern, 3008 Bern, Switzerland; tim.vanbellingen@unibe.ch; 8Department of Orthopaedic Surgery and Rehabilitation, Wake Forest School of Medicine, Winston-Salem, NC 27157, USA

**Keywords:** BDNF, exercise, Parkinson’s disease

## Abstract

Introduction: Exercise therapy may increase brain-derived neurotrophic factor (BDNF) levels and improve clinical outcomes in people living with Parkinson’s disease (PD). This systematic review was performed to investigate the effect of exercise therapy on BDNF levels and clinical outcomes in human PD and to discuss mechanisms proposed by authors. Method: A search on the literature was performed on PubMed up to December 2023 using the following key words: Parkinson’s disease AND exercise, exercise therapy, neurological rehabilitation AND brain-derived neurotrophic factor, brain-derived neurotrophic factor/blood, brain-derived neurotrophic factor/cerebrospinal fluid AND randomized clinical trial, intervention study. Only randomized clinical trials comparing an exercise intervention to treatment as usual, usual care (UC), sham intervention, or no intervention were included. Results: A meta-analysis of BDNF outcomes with pooled data from five trials (N = 216 participants) resulted in a significant standardized mean difference (SMD) of 1.20 [95% CI 0.53 to 1.87; Z = 3.52, *p* = 0.0004, I^2^ = 77%], favoring exercise using motorized treadmill, Speedflex machine, rowing machine, and non-specified exercise. Significant improvements were found in Unified Parkinson’s Disease Rating Scale (UPDRS), UPDRS-III, 6 Minute Walk Test (6MWT), and Berg Balance Scale (BBS). Methodological quality of trials was categorized as “good” in three trials, “fair” in one trial, and “poor” in one trial. Conclusion: Key results of this systematic review are that exercise therapy is effective in raising serum BDNF levels and seems effective in alleviating PD motor symptoms. Exercise therapy confers neuroplastic effects on Parkinson brain, mediated, in part, by BDNF.

## 1. Introduction

Parkinson’s disease (PD) is a complex, chronic, highly disabling neurodegenerative condition and the fastest accelerating brain disease in the world. Despite extensive research on PD, a cure has yet to be discovered, underscoring the complex and multifaceted nature of this neurodegenerative disorder. Although the cause of PD is unknown, it has been linked to certain genetic factors, as well as modifiable risk factors, including environmental factors (e.g., toxin exposure [1]), and lack of exercise or low levels of physical activity [2]. The prevalence of PD has doubled in the last 25 years to 7 million globally in 2019 and is projected to affect 30 million people of all ages worldwide by 2040 [3]. Yang et al. [4] approximated the economic burden of PD in the United States to be around USD 51.9 billion and projected it to surpass USD 79 billion by 2037. Parkinson’s disease causes a wide variety of non-motor features (NMF) and motor symptoms including tremor, dystonia, rigidity, bradykinesia, as well as problems with gait, motor coordination, and balance [5]. Non-motor features include behavioral changes, depression and anxiety, cognitive impairment (specifically relating to learning and memory), constipation, osteoporosis, pain, insomnia, and fatigue [6]. The medical management of these issues can be ameliorated to a certain extent pharmacologically with L-dopa or, in the later stages, with neurosurgical approaches such as deep brain stimulation. Currently, the medical management of PD involves dopamine replacement therapy (DRT). DRT is symptomatic, alleviating certain dopa-responsive motor symptoms such as tremor and bradykinesia, but it fails to address many gait and non-motor mobility-related problems. Furthermore, highly disabling long-term side effects may arise from DRT including dyskinesia and motor fluctuations [7]. The growing prevalence, severity, economic burden, and lack of a cure for PD emphasize the need for effective alternative treatments. Recently the World Health Organization articulated the need for actionable priorities to increase the quality of care of people living with PD [8], including self-management through exercise.

Historically, the physiologic use of exercise was believed to be limited or even harmful to people living with PD, as exercise might increase muscle tone, exacerbating rigidity. Today, a rich vein of evidence supports the use of exercise and physiotherapy, beginning at diagnosis in persons with PD who do not display contraindications to exercise [9,10,11]. Systematic reviews of randomized controlled trials (RCT) have shown that comprehensive rehabilitation, including aerobic exercise, resistance training, gait and balance training, stretching, community-based training, and cueing, improves both motor symptoms and non-motor features, physical performance, general health and wellbeing, and quality of life among individuals at all stages of PD [12,13,14,15,16,17,18].

In animal and human models, the physiologic use of exercise leads to the expression of genes related to improved neuronal proliferation, survival, and reduced inflammatory response [19,20,21,22]. Exercise upregulates neurotrophins including nerve growth factor, glia-derived neurotrophic factor (GDNF), production of GDNF-producing cells (glia), and brain-derived neurotrophic factor (BDNF)-induced tropomyosin receptor kinase B (TrkB) signaling in lymphocytes [23]. Exercise prevents the downregulation of the BDNF signaling pathways in the substantia nigra and striatum [24,25,26,27]. This, in turn, exerts cytoprotective effects on dopaminergic neurons in the basal ganglia [28,29] and other areas of the brain including the bed nucleus of the stria terminalis, septum, cerebellum (nucleus of solitary tract), dentate gyrus of the hippocampus, and cerebral cortex [30,31,32,33,34]. BDNF regulates the amount of dopamine and dopaminergic cell activity in the striatum [35], and, in turn, dopaminergic input regulates the sensitivity of striatal spiny neurons to BDNF [36]. People with PD exhibit decreased levels of BDNF in the nigrostriatal pathway, compared to neurotypical age-matched controls [37,38], which may leave the brain more vulnerable to degeneration [19,39]. It has been hypothesized that exercise-induced increase in circulating BDNF levels may serve as a therapeutic agent in PD [20].

Early studies on the effect of voluntary exercise on BDNF levels in PD were conducted by Zoladz et al. [40] and Marusiak et al. [41], using a pretest–posttest design. Zoladz and Marusiak evaluated the effect of stationary cycle ergometer interval training in a motor control laboratory (three times per week for one hour each session, for eight weeks) on BDNF serum levels in eleven outpatients with PD (Hoehn and Yahr stages 1.5–3) and neurotypical controls. They found alleviated parkinsonian hypertonia and myometric stiffness in the biceps brachii accompanied by increased basal serum BDNF levels. Using an uncontrolled pretest–posttest design, Angelucci and colleagues [42] evaluated serum BDNF levels among nine adults with rigid-akinetic PD (Hoehn and Yahr stages 2–3) after inpatient rehabilitation (three times per day, five days per week, for four weeks of intervention consisting of cycling, physical therapy, treadmill training, and Wii Fit balance board training). Angelucci’s intervention resulted in improved clinical measures (UPDRS-part II and III (gait and balance score) and quality of life) and was accompanied by increased BDNF levels after one week of training.

The first meta-analysis evaluating the effects of exercise on BDNF in human PD showed a significant summary effect size for BDNF in two trials [43]. In 2020, Johansson et al. [44] included three studies without control groups in their meta-analysis of exercise effects on BDNF levels, showing no significant summary effect size. In 2023, Li et al. [45] and Rotondo and colleagues [46] pooled data on exercise studies, showing positive effects on BDNF levels compared to controls. Due to the conflicting meta-analytic results and the lack of human PD trials detailing mechanisms of exercise on BDNF levels, the purpose of the present systematic review and meta-analysis is to give an updated overview of the mechanisms and effects of exercise therapy on BDNF levels in PD, in order to bring these insights into the clinical context of physical medicine and rehabilitation for people living with PD.

## 2. Method

### 2.1. Data Sources and Search Strategy

This study was conducted following the Preferred Reporting Items for Systematic Reviews and Meta-Analyses (PRISMA) statement [47] and was not registered in any database. A systematic literature search was conducted on PubMed, Embase, and Scopus until 1 December 2023. We included the following key words: Parkinson’s disease AND exercise, exercise therapy, neurological rehabilitation AND brain-derived neurotrophic factor, brain-derived neurotrophic factor/blood, brain-derived neurotrophic factor/cerebrospinal fluid AND clinical trial, intervention study. The final search was formulated as follows: “(“Parkinson Disease” [Mesh] OR Parkinson disease [tiab] OR Idiopathic Parkinson’s disease [tiab] OR Lewy body Parkinson’s disease [tiab] OR Primary Parkinsonism [tiab] OR Parkinson [tiab]) AND (“Brain-Derived Neurotrophic Factor” [Mesh] OR “Brain-Derived Neurotrophic Factor/blood” [Mesh] OR “Brain-Derived Neurotrophic Factor/cerebrospinal fluid” [Mesh] OR Nerve Growth Factors [Mesh] OR Brain Derived Neurotrophic Factor [tiab] OR BDNF [tiab] OR Neurotrop* [tiab]) AND (“Exercise” [Mesh] OR “Exercise Therapy” [Mesh] OR “Neurological rehabilitation” [Mesh] OR Exercise* [tiab] OR Physical exercise* [tiab] OR Physical activit* [tiab] OR Aerobic exercise* [tiab] OR Training [tiab] OR Therapy exercise* [tiab] OR Rehabilitation exercise* [tiab] OR Neurorehabilitation [tiab] OR Neurologic rehabilitation [tiab]) AND (“Clinical trial” [Mesh] OR Intervention study [tiab] OR Randomized Controlled Trial [tiab] OR Randomised Controlled Trial [tiab] OR RCT [tiab] OR Randomi* [tiab] OR Pilot study [tiab])”.

In addition to a systematic literature search, papers were also sought through reference tracking, and a free text search with the aforementioned terms was conducted.

### 2.2. Criteria for Inclusion

Inclusion criteria were as follows: (1) human participants with PD diagnosis; (2) the study is a randomized controlled trial; (3) at least one of the trial arms consists of a control condition which can be either usual care (UC), treatment as usual, sham intervention, or no intervention; (4) the trial contains an exercise component; (5) plasma or serum BDNF is measured before and after the intervention; (6) article is written in English, German, or Dutch.

### 2.3. Data Extraction

Data from relevant trials were extracted directly from the text, tables, and figures. Data were extracted on number of participants, and participant age, sex, disease duration, and disease severity (in relation to motor symptoms and non-motor features of PD) were measured with the Hoehn and Yahr scale, the gold standard, and the Movement Disorder Society—Unified Parkinson’s Disease Rating Scale (MDS-UPDRS) part-III score. Extracted data about the trial protocol included exercise dosage (i.e., duration of each session, number of sessions per week, and total amount of weeks) and exercise type and intensity (as measured with Maximum Heart Rate (HR_max_) or Heart Rate Reserve (HRR) when available). Relevant outcome data were extracted pre and post intervention as mean and standard deviation (SD) when available. If not available, they were approximated using the method reported by Wan et al. [48]. The table and formulas used are provided in Table 1. The extracted data included blood and serum BDNF levels in picograms per milliliter (pg/mL) and several clinical measures including the UPDRS, UPDRS-III, six-minute walk test (6MWT), Berg Balance Scale (BBS), Tinetti’s test for balance, and the Parkinson’s fatigue scale (PFS-16). The proposed mechanisms described by authors in their trial on how BDNF levels change with exercise were also extracted when available. The authors were contacted about missing data or when clarification about their data was needed.

### 2.4. Effect Size Analysis

A random-effects meta-analysis was conducted using Review Manager [49] for the outcome measures BDNF and UPDRS-III. The pooled results were reported as standardized mean difference (SMD) with 95% confidence intervals (CI). Heterogeneity was evaluated using the I^2^ statistics. When appropriate, we performed sensitivity analysis through the elimination of specific studies or comparisons.

### 2.5. Quality Assessment

The methodological quality of individual studies was appraised using the National Institutes of Health (NIH) study-quality assessment tools [50]. This tool includes items for evaluating the internal quality of trials. Trials were categorized as “good” quality when more than 80% of the criteria were met, “fair” quality when 50–80% of the criteria were met, and “poor” quality when less than 50% of the criteria were met or when a substantial risk of bias was present. An overview of the criteria is provided in Table 2.

## 3. Results

### 3.1. Selection of Studies

The search strategy resulted in a yield of 27 unique studies. The title/abstract screening resulted in ten eligible studies, of which three met the aforementioned inclusion criteria. Nine more studies were identified via reference tracking, of which two met the inclusion criteria. Figure 1 displays the selection process according to the PRISMA guidelines.

### 3.2. Quality Assessment

The quality of all trials was assessed. Three studies [51,52,53] were categorized as “good”, one trial [54] was categorized as “fair”, and one trial was categorized as “poor” [55]. Table 2 shows individual results for each of the 14 NIH quality criteria questions used to assess the methodological quality and make an informed decision about the possible risk for bias of each of the included trials, and it provides numerical results to assess the total methodological quality score and rating. In addition, this table provides assessment results of the effect size (SMD [range]) and total dosage for each of the included trials.

### 3.3. Summary of the Literature

The population characteristics, trial design, exercise protocol, and results are described in Table 3. The trials were published between 2013 and 2023, and a total of 216 participants with PD contributed to the articles included in this systematic review. Three trials [52,54,55] conducted a moderate-intensity exercise program, and two trials [51,53] conducted a vigorous-intensity exercise program according to standards for exercise intensity articulated by Liguori and the American College of Sports Medicine (ACSM) [56]. Additionally, exercise frequencies ranged from two to five sessions per week, and intervention duration ranged from 20 to 180 min per session over a period of four to twelve weeks. Furthermore, the exercise mode was via a rowing machine [51], via a motorized treadmill [52,53], via a Speedflex machine [53], or without a specified exercise aid [51,54,55].

All five trials reported serum concentrations of BDNF [51,52,53,54,55]. In addition, BDNF was measured through an enzyme-linked immunosorbent assay (ELISA) kit [51,52,53,54,55]. Three trials [51,52,55] reported clinical outcomes as an outcome measure. Outcome measures for clinical outcomes included UPDRS [52,55], UPDRS-III [52,55], 6MWT [51,52], BBS [52], Tinetti’s test for balance [51], and PFS-16 [51]. One trial [52] reported on correlations between BDNF and clinical outcome measures. DiCagno et al. [51] was the only study that explicitly reported on patient and public involvement (PPI) in the study process. None of the other included trials mentioned PPI.

### 3.4. BDNF Changes

Four trials [51,52,53,54] reported a statistically significant increase in serum BDNF levels. DiCagno et al. [51] found a statistically significant inter-group difference in BDNF between the low-frequency group (LFG) and control group (CG). No significant pre-post changes were reported. Frazzitta et al. [52] found an increase in serum BDNF of 14.46% (*p* = 0.017) in the IRT group and reported a statistically non-significant change in serum BDNF in the UC group. Inter-group differences were statistically significant (*p* = 0.0017). O’Callaghan et al. [53] reported a statistically non-significant change in the moderate-intensity continuous training group (*p* = 0.650) and a significant increase in serum BDNF of 5.55% (*p* = 0.010) in the high-intensity interval training group. The change in both control groups was not statistically significant. Szymura et al. [54] found an increase in serum BDNF of 43.3% (*p* = 0.011) in the balance training group with PD. Freidle et al. [55] reported no statistically significant post-intervention changes in serum levels (*p* > 0.05).

### 3.5. Clinical Outcomes

Two trials [51,52] reported significant changes in clinical outcomes. DiCagno et al. [51] found an increase in 6MWT score of 35.16% (*p* < 0.05) in the LFG. Inter group differences between the LFG and CG compared to the HFG and the CG compared to the LFG were significant (*p* < 0.05). DiCagno et al. [51] also found an increase in Tinetti’s balance score of 30.23% (*p* < 0.05) and a decrease of 55.73% (*p* < 0.05) in the LFG. Frazzitta et al. [52] found a significant decrease in total UPDRS and UPDRS-III score of 10.64 (*p* < 0.0001) and 7.6 (*p* < 0.0001), respectively, in the IRT group and a statistically non-significant change in UPDRS-III score in the UC group (*p* = 0.1934). Inter-group differences were statistically significant (*p* < 0.001). Frazzitta et al. [52] also found an increase in 6MWT of 24.54% (*p* = 0.0001) and an increase in BBS score of 5.36 (*p* = 0.0016) in the IRT group. Freidle et al. [55] reported no statistically significant changes in total UPDRS and UPDRS-III outcomes.

### 3.6. BDNF–Clinical Outcomes Correlation

One trial [52] investigated possible correlations between BDNF and clinical outcome measures. Frazzitta et al. [52] found no significant correlation between BDNF level change and changes in UPDRS-III (r = −0.13; *p* = 0.65), UPDRS (r = −0.18; *p* = 0.52), 6MWT (r = 0.05; *p* = 0.88), and BBS (r = −0.11; *p* = 0.69).

### 3.7. Meta-Analysis

BDNF levels assessed with laboratory measures were reported in seven comparisons of the five RCTs [51,52,53,54,55] (*n* = 216), and pooling resulted in a significant heterogeneous SES (SMD 1.20 [95% CI 0.53 to 1.87; Z = 3.52, *p* = 0.0004, I^2^ = 77%, Figure 2a). 

The MDS-UPDRS motor score was reported in two RCT’s [52,55] (N = 119), and pooling resulted in a statistically non-significant heterogeneous SES (SMD −3.38 [95% CI −10.94 to 4.19; Z = 0.88, *p* = 0.38, I^2^ = 97%, Figure 2b).

When eliminating Szymura et al. [54] from the meta-analysis, heterogeneity dropped substantially, and the SES dropped to 0.79 (95% CI 0.49 to 1.08; Z = 5.21; *p* < 0.00001; I^2^ = 0%, Figure 2c). 

When eliminating DiCagno et al. [51] from the meta-analysis, heterogeneity rose, and the SES increased to 1.41 (95% CI 0.49 to 2.33; Z = 3.00; *p* < 0.0001; I^2^ = 84%, Figure 2d).

## 4. Discussion 

This present systematic review was performed to investigate whether exercise therapy has an effect on BDNF levels and clinical outcomes in human PD. The main findings are that the pooling of the five available RCTs produced a significant SES of 1.20 [95% CI 0.53, 1.87] in favor of a positive exercise effect on serum BDNF levels (Table 3 and Figure 2a).

We performed a sensitivity analysis through the elimination of specific studies or comparisons and found the following curiosities:

When we eliminate Szymura [54] from the meta-analysis (Figure 2c), heterogeneity drops substantially, suggesting that the Szymura trial contributes to most of the heterogeneity amongst the RCTs included in the present meta-analysis. This is plausible given our result that Szymura has a much higher SMD when compared to other included RCTs. We identified no other factors that could have contributed to the heterogeneity (e.g., age, sex, disease severity). The limited dataset did not allow for a formal analysis of exercise intervention dosage, and Table 2 did not reveal a clear trend for higher effect sizes with increased exercise dosage. When we excluded DiCagno [51] from the meta-analysis (Figure 2d), we found that the SES increased to 1.41 and stayed statistically significant. We investigated the contribution of DiCagno to the total SES because DiCagno used a novel multimodal exercise regime with electromyostimulation. We hypothesized that the effect of exercise on BDNF could have been attributed more to the electromyostimulation than the exercise itself, which would distort our results. However, our SES even increased when DiCagno was excluded from the analysis. Based on this result, we propose that electromyostimulation did not severely affect our meta-analytic result. The elimination of other studies did not result in a drastic change in SES or heterogeneity.

To the best of our knowledge, the current meta-analysis is the largest (N = 216) to-date of exercise effects on BDNF levels in human PD. Prior meta-analytic studies have reported inconsistent results (see Appendix A). In 2020, Johansson et al. [44] reported meta-analytic results from 35 participants with PD, pooling results from one randomized controlled trial [52] and two uncontrolled pre-post studies [40,42]. Overall, this methodology led to a statistically non-significant heterogeneous summary effect size for the effects of exercise on BDNF concentration. It is plausible that Johansson’s variable results could be attributable to a number of factors, including pooling results from randomized and non-randomized trials in the meta-analysis or the pooling of experimental values in the analysis without including corresponding values from a control group. 

In 2023, Li and colleagues [45] reported results of their meta-analysis pooling data on the exercise-induced effects on BDNF concentration from five comparisons in four RCTs with 192 participants with PD [52,53,54,55]. Overall, their meta-analytic procedure led to a statistically significant result of the effects of exercise versus control on BDNF concentration (see Appendix A). These results were confirmed in the recent meta-analysis by Rotondo and colleagues [46] who pooled data on exercise-induced effects on BDNF concentration from four comparisons in three RCTs with 180 participants with PD [52,54,55]. Overall, pooling the data showed a statistically significant beneficial effect of the effects of exercise versus control on BDNF concentration in PD (see Appendix A).

Several other non-controlled studies, due to lack of proper control not included in our meta-analysis, support the notion that exercise therapy has positive effects on BDNF and various clinical outcomes [33,34,40,41,42,58,59,60,61]. Recently, Gomes et al. [62] also showed promising results by studying the effects of High-Intensity Interval Training (HIIT) on BDNF and motor and non-motor outcomes using a single-case experimental design. HIIT is thought to be an efficient, effective, and more motivating alternative to continuous aerobic exercise; however, future controlled studies should investigate further whether HIIT induces neuroplastic effects. 

In this systematic review, quality evaluation was achieved via a valid measure for assessing quality and risk of bias, the National Institutes of Health Quality Assessment Tool (NIH QAT) [50]. The NIH QAT has been used extensively internationally since 2017, including in systematic reviews of trials involving rehabilitation populations. Therefore, its use is felt to be justifiable in the present systematic review.

Patient and public involvement (PPI) was only mentioned explicitly in one study [51]. In the future it would be important to determine whether PPI affects the outcomes of trials and has significant effect on the methodological quality or effect size of the studies.

Overall, based on the results from meta-analyses and systematic reviews to-date, it seems highly plausible that exercise has a positive effect on blood-based BDNF levels. Most studies have reported an increase in BDNF following exercise as opposed to their UC or non-exercise control groups. However, two questions remain unclear: (1) what exercise intensity is most effective in increasing BDNF levels? And (2) what exact mechanisms offer increased exercise-induced circulating BDNF levels in human PD? 

The results from O’Callaghan et al. [53] seem to suggest that high-intensity exercise is more effective in enhancing BDNF, while DiCagno et al. [51] only found a significant effect on BDNF in its low-frequency group. This difference in findings could have a number of causes, e.g., differences in study population characteristics, trial design, and exercise modality and/or dosing, but due to the limited number of papers comparing exercise modalities, no formal analysis was conducted. Although Frazzitta et al. [52] found a significant improvement in UPDRS score, pooling with Freidle et al. [55] resulted in a non-significant change in UPDRS score. The analysis also showed a high heterogeneity between the studies, which could be attributed to differences in participant characteristics, exercise modality, and exercise dosage between the studies. 

With exercise, people with PD show a remarkable ability to overcome their motor deficits. Frazzitta et al. [52] reported significant changes in BBS, 6MWT, UPDRS-II, and UPDRS-III, and DiCagno et al. [51] reported significant changes in the 6MWT, Tinetti’s test for balance, and Parkinson Fatigue Scale, but Freidle et al. [55] reported no significant change in UPDRS and UPDRS-III scores. Although the evidence is conflicting, it is plausible that moderate- to vigorous-intensity exercises can improve distance walked and equilibrium, and also alleviate clinical motor symptoms. 

### Exercise Neuroplastic Mechanisms

Candidate exercise-induced neuroplastic mechanisms include altered functional connectivity, corticomotor excitability, D2 and D3 dopamine receptor availability, endogenous release of anti-inflammatory cytokines, increased dorsal striatal dopamine release, volumetric changes, and increased neurotrophin level. 

Pineda-Pardo [63] showed that striatal dopaminergic denervation in early PD follows a somatotopic pattern, beginning with the upper limb representation in the putamen. The changes in the putamen correlated with the evolution of motor features. In the 6-month Park-in-Shape, home-based, remotely supervised aerobic exercise trial by van der Kolk [64], individuals with mild to moderate PD experienced changes in dorsolateral prefrontal cortex connectivity within the right frontoparietal networks proportional to fitness improvements, and a posterior-to-anterior shift in the balance of corticostriatal sensorimotor connectivity [65]. These results were interpreted by the authors as stabilizing motor disease progression and enhancing cognition by stimulating functional and structural plasticity in corticostriatal sensorimotor and cognitive control networks [65] (p. 215). In the study by Sehm et al. [66], changes in the cortical and subcortical volumes of structurally connected lateralized parietal–basal ganglia circuitry networks in the left inferior parietal cortex and right lingual gyrus were found with balance training exercise (parietal–basal ganglia circuitry deficits correlate with impaired set shifting and dual-task walking in PD). In the study by Fisher et al. [67], the increase in corticomotor excitability was observed after two months of intense treadmill training with or without the use of a partial body weight support harness in patients with early-stage PD (low excitability is a marker for PD severity). In a follow-up study, Fisher et al. [68] showed that two months of motorized treadmill training increased dopamine D2 type receptor-binding potential in the basal ganglia of patients with early-stage unmedicated PD. Sacheli [69] showed that three months of aerobic cycle training increased ventral striatal dopamine release proportional to fitness improvements in patients with mild-to-moderate PD. Malczynska and colleagues [70] showed that, in PD, three months of high-intensity interval training exercise increased the synthesis of anti-inflammatory cytokines. The emerging neuroscience evidence described above hints that exercise therapy induces complex adaptive neuroplasticity changes in cortical and basal ganglia circuitries [21,22,71,72,73] 

Some authors of the trials included in the present meta-analysis provided vague rationales for BDNF mechanisms (Table 3), including release of intracellular BDNF, secondary to HIIT-associated muscle injuries [54], and exercise-induced, transient hypoxia [53], while no mechanistic explanation was provided by Dicagno et al. [51], Frazzitta et al. [52], and Freidle et al. [55]. Possible sources of BDNF include brain/neurons, skeletal muscle, mononuclear cells, platelets, and vascular endothelial cells. The brain, and specifically the hippocampus, is thought to be a key source of BDNF at rest and at times of increased metabolic demand such as exercise [74,75,76]. Schaeffer et al. [34] showed an increase in serum BDNF in healthy controls and people with PD, but this increase failed to reach statistical significance. However, this study showed a significant increase in the volume of certain left hippocampal subsections in the PD group after a six-week exergaming intervention. Fernandes et al. [77] argued that exercise impacts the epigenetic regulation of BDNF gene expression in neurons. They described a mechanism in which exercise activates N-methyl-D-aspartate (NMDA) receptors, triggering an influx of calcium into neurons, which, through a cascade of molecular events, results in the activation of cAMP response elements (CREs)-binding protein and transcription of BDNF. Certain exercise byproducts facilitate this molecular pathway and others, including reactive oxygen species (produced as a result of higher mitochondrial activity [78]), circulating lactate [79], tumor necrosis factor alpha (TNF-α) [80], hypoxia-inducible factor-1 [81], and b-hydroxybutyrate [82], as promoters of exercise-induced BDNF transcription and translation. 

It is plausible that skeletal muscle may both directly and indirectly increase circulating BDNF. Myokines irisin and cathepsin B, which are released when skeletal muscle contracts, play a role in the regulation of BDNF production in the hippocampus [83]. Skeletal muscle is also a major source of lactate, which may cross the blood–brain barrier and enhance BDNF production [84]. It is also plausible that hematopoietic cells play a role in BDNF production. One study demonstrated that mononuclear cells express BDNF during exercise in a dose-dependent manner [85], although the significance of this contribution is not fully understood. Platelets contain 99% of total blood-borne BDNF, although their primary role is thought to be more related to the storage of BDNF than production [86]. It is postulated that megakaryocytes produce the majority of BDNF found in platelets, and the remainder is sequestered [87]. However, platelets do play an important role in the serum BDNF concentration as they release BDNF in a dose-dependent response after shear stress, such as being induced by exercise [88]. Finally, it is plausible that vascular endothelial cells play a role in BDNF production. Vascular endothelial cells in both the central and peripheral nervous systems produce, store, and release BDNF in response to shear stress and hypoxia [89]. One study found that cerebral endothelial cells produce fifty times more BDNF than neurons, indicating that endothelial cells may be responsible for a significant portion of BDNF in the brain [90]. In support of this theory, another study found that the removal of cerebral endothelium in animal models led to a significant decrease in BDNF content in the brain [91]. 

In summary, while the brain remains the primary source of BDNF during exercise, it is important to recognize the contribution of peripheral sources. The role of BDNF in exercise-induced plasticity in PD is not fully understood [92,93]. By extrapolating insights from animal and human studies on BDNF production during exercise, we may better understand future targets for research and potential therapeutic interventions to further harness the benefits of BDNF in PD. 

## 5. Limitations 

There are limitations in this systematic review. Firstly, this review is likely susceptible to publication bias, where interventions with an effect are more likely to be published than interventions without an effect, which could have distorted our results. Secondly, most of the included studies had very small sample sizes. Thirdly, some data had to be estimated due to means and SDs not being reported in the DiCagno and O’Callaghan trials. Finally, although our findings support an association between exercise and BDNF levels in PD, large, long-term (1–2 years follow-up), rigorously conducted race- and sex-diverse studies with more homogeneous samples of persons “on” and “off” medication at all stages of Parkinson disease with and without non-motor impairment are needed to evaluate the efficacy and effectiveness of exercise interventions and their generalizability, including the impact on motor and non-motor symptoms and quality of life as assessed with the MDS-UPDRS. In addition, only a few PD exercise intervention trials have focused in-depth on diversity issues of heterogeneity such as the impact of ethnicity or genetic factors on clinical effects of exercise and BDNF outcome. These factors, in addition to others such as variations in participant characteristics, study design, intervention protocols, and outcome measures, could be important sources of heterogeneity in meta-analyses that, if taken into account, could lead to new insights.

## 6. Future Directions

The results of this meta-analysis show promising evidence of blood-based BDNF concentration increasing through exercise in PD. However, current studies report a BDNF increase directly after the intervention but lack a longer-term follow-up period. Likewise, there is currently not enough evidence to state the most optimal exercise regime (i.e., exercise type, dosage, and intensity), and our current understanding of the mechanisms behind and the effects of BDNF increase on clinical outcomes remains unclear. Future research should, therefore, consist of thorough and high-quality randomized controlled trials to further investigate the effect of exercise on BDNF levels, mechanisms of exercise effect, and clinical outcomes in PD. Trials should involve more participants, more actively involve participants in trial design and execution, last longer, and work towards a personalized exercise protocol prescription for PD patients due to the heterogeneity in PD symptoms. Moreover, research directly comparing various types and intensities of exercise with a delay start or waiting list setup should be performed to determine the most effective exercise regime. 

The exercise trials discussed above measured BDNF using validated commercial ELISA kits. Blood draws are quite invasive and limited to clinical settings. These limitations could increase participant drop-outs in studies in which BDNF is sampled frequently. Future studies ought to evaluate the reliability of non-invasive BDNF measures, such as in saliva, and response to exercise intervention [94,95]. Finally, future exercise studies ought to evaluate whether BDNF could be a viable marker in early stages/prodromal PD and the utility of exercise-induced effects on BDNF levels in tracking disease progression. The field of exercise trials and BDNF plasticity is still in its very early infancy. Future trials will shed additional light on neuroplasticity–neurotrophin mechanisms. More research is warranted. 

This systematic review shows exercise to be beneficial by attenuating motor symptoms and possibly by promoting BDNF-mediated neuroplastic effects in PD. Personalized exercise programs are feasible, accessible, and safe, as shown by Harpham et al. [96] in a high-intensity interval training protocol in the home environment. A recent study [97] conducted by experts in multidisciplinary rehabilitation and PD found rehabilitation to be underutilized especially in early-stage PD and defined key principles for rehabilitative care. This, combined with our findings, could lead to PD treatment strategies shifting from more medication-focused towards exercise-focused, and we could see personalized exercise regiments becoming a more valuable tool for combating motor symptoms, especially in early PD. Furthermore, the role of physiotherapists in treating PD might move from maintaining functional ability, individuality, and minimizing risk of falls to being more intertwined with facilitating exercise in PD patients, possibly progressively in the home environment, making exercise therapy more convenient for PD patients. However, more concrete evidence about exercise effects, optimal exercise mode, and a further understanding of the mechanisms are indispensable for integrating exercise into PD treatment.

Although BDNF has been established as a promising biomarker in the development, survival, and function of corticostriatal cells, with plausible response to physical exercise in PD, the evidence supporting the mechanisms is still insufficient and not well understood, and, thus, practical applications of BDNF leading to a possible long-term protection, improvement, or permanent rescue of brain in PD are still speculative. We are encouraged by a rigorously conducted, large-scale, ongoing international exercise neuroplasticity trial, the SPARXS3, conducted by Patterson, Corcos, and colleagues (28 sites in USA and Canada https://www.sparx3pd.com (accessed on 4 January 2024)). SPARXS3 enrolls adults with recent PD (<3 years) in motorized treadmill walking exercise to evaluate clinical and neuroprotective effects including BDNF at 12, 18, and 24 months post exercise [98]. The SPARXS3 study is funded by the National Institutes of Health and will advance knowledge about clinical effects of exercise on PD motor and non-motor features and concomitant neuroplasticity.

A recent review by Cammisuli et al. [99] showed promising neuroprotective effects in studies conducted on the combination of diet and aerobic exercise in PD. Nutrition, therefore, might be an important component in facilitating neuroprotective effects in PD, and the combination of diet and exercise might be an exciting area for future research, especially regarding the molecular mechanisms and optimal diet regime.

## 7. Conclusions

This systematic review shows that exercise therapy is effective in raising blood-based BDNF levels in patients with PD and that physical exercise might be a useful strategy in the treatment of PD symptoms. However, longer-term effects, optimal exercise modality, and the mechanisms of exercise in increasing BDNF levels are yet to be determined.

## Figures and Tables

**Figure 1 brainsci-14-00194-f001:**
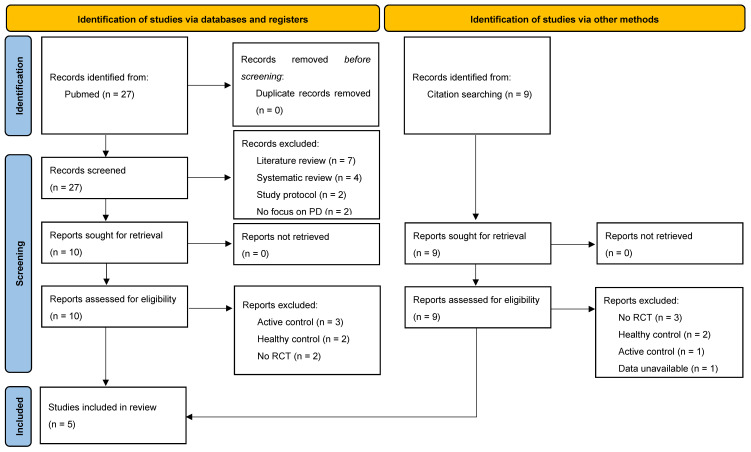
PRISMA flow diagram of selection process.

**Figure 2 brainsci-14-00194-f002:**
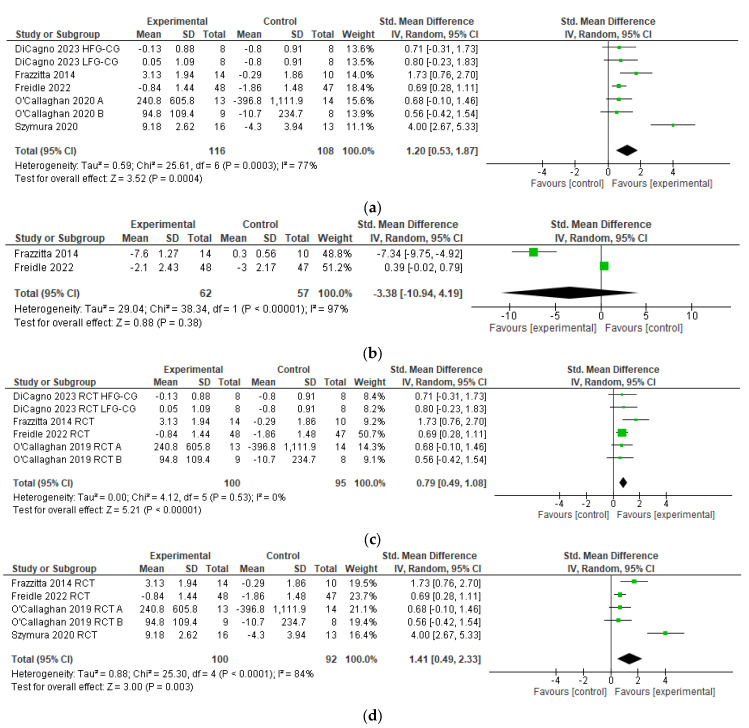
(**a**) Summary effect sizes for outcome of change in serum BDNF levels [51,52,53,54,55]. (**b**) Summary effect sizes for outcome of change in MDS-UPDRS motor score. (**c**) Summary effect sizes for outcome of change in serum BDNF levels excluding Szymura 2020 [54]. (**d**) Summary effect sizes for outcome of change in serum BDNF levels excluding DiCagno 2023 [51]. (**a**–**d**) Green squares indicate individual SES. Black colored diamonds indicate the summary effect size; HFG = high-frequency group, LFG = low-frequency group, CG = control group, SD = standard deviation, Std = standardized, IV = inverse variance, CI = confidence interval, Tau^2^ = statistic to determine true variance, Chi^2^ = chi-square statistic, df = degrees of freedom, P = *p*-value, I^2^ = statistic to determine heterogeneity, Z = z-score.

**Table 1 brainsci-14-00194-t001:** Formulas and table used for estimating means and SDs from Wan et al. [48].

$\bar{X}\approx \frac{q_{1}+m+q_{3}}{3}$									
$S\approx \frac{q_{3}-q_{1}}{\eta \left(n\right)}$									
n=4Q+1									
** * Q * **	** * η * ** ** (*n*) **	** * Q * **	** * η * ** ** (*n*) **	** * Q * **	** * η * ** ** (*n*) **	** * Q * **	** * η * ** ** (*n*) **	** * Q * **	** * η * ** ** (*n*) **
1	0.990	11	1.307	21	1.327	31	1.334	41	1.338
2	1.144	12	1.311	22	1.328	32	1.334	42	1.338
3	1.206	13	1.313	23	1.329	33	1.335	43	1.338
4	1.239	14	1.316	24	1.330	34	1.335	44	1.338
5	1.260	15	1.318	25	1.330	35	1.336	45	1.339
6	1.274	16	1.320	26	1.331	36	1.336	46	1.339
7	1.284	17	1.322	27	1.332	37	1.336	47	1.339
8	1.292	18	1.323	28	1.332	38	1.337	48	1.339
9	1.298	19	1.324	29	1.333	39	1.337	49	1.339
10	1.303	20	1.326	30	1.333	40	1.337	50	1.340

$\bar{X}$ = Mean, *S* = standard deviation, *q*_1_ = 1st quartile, *q*_3_ = 3rd quartile, m = median, *η* (*n*) = a function of *n*, *Q* = a positive integer between 1 and 50.

**Table 2 brainsci-14-00194-t002:** Results of the quality assessment.

Quality Criteria	DiCagno	Frazzitta	Freidle	O’Callaghan	Szymura
#1	+	+	+	+	+
#2	+	+	+	+	CD
#3	+	+	+	+	NR
#4	-	-	-	-	-
#5	+	+	+	+	-
#6	+	+	+	+	+
#7*	+	+	-	+	+
#8*	+	+	-	+	+
#9	NR	NR	-	+	NR
#10	+	+	+	+	+
#11	+	+	+	+	+
#12	+	+	+	-	-
#13	+	+	+	+	+
#14*	+	+	+	+	+
Total score	12/14	12/14	10/14	12/14	8/14
Rating	good	good	Poor *	good	fair
Effect size (SMD [range])	0.71 [−0.31–1.73] 0.80 [−0.23–1.83]	1.73 [0.76–2.70]	0.69 [1.28–1.11]	0.68 [−0.10–1.46] 0.56 [−0.42–1.54]	4.00 [2.67–5.33]
PPI	yes	no	no	no	no
Dosage	480	3600	1200	2160	2160

Abbreviations: + = criterion is met, - = criterion is not met, CD = cannot determine, NR = not reported, PPI = patient and public involvement, SMD = standardized mean difference. Studies are categorized as “good” quality when more than 80% of criteria are met, “fair” quality when 50–80% of criteria are met, and “poor” quality when less than 50% of criteria are met. #1 Was the study described as randomized, a randomized trial, a randomized clinical trial, or an RCT? #2 Was the method of randomization adequate? #3 Was the treatment allocation concealed? #4 Were study participants and providers blinded to treatment group assignment? #5 Were the people assessing the outcomes blinded to participants’ group assignment? #6 Were the groups similar at baseline on important characteristics that could affect outcomes? #7* Was the overall drop-out rate from the study at endpoint 20% or lower of the number allocated to treatment? #8* Was the differential drop-out rate at endpoint 15% or lower? #9 Was there high adherence to the intervention protocols for each treatment group? #10 Were other interventions avoided or similar in the groups? #11 Were outcomes assessed using valid and reliable measures, implemented consistently across all study participants? #12 Did the authors report that the sample size was sufficiently large to be able to detect a difference in the main outcome between groups with at least 80% power? #13 Were outcomes reported or subgroups analyzed prespecified? #14* Were all randomized participants analyzed in the group to which they were originally assigned? * Marks issue with methodological quality on items 7, 8, or 14, i.e., high drop-out rate. These items mark a significant risk for bias. Dosage is calculated as follows: minutes/session×sessions/week×number of weeks. It should be noted that the blinding of the study participants and providers is impossible due to the nature of the intervention.

**Table 3 brainsci-14-00194-t003:** Summary of included randomized controlled trials [51,52,53,54,55].

Author (Year)	Intervention	Control	Protocol/Exercise Components	Results	Proposed Mechanism
DiCagno et al. (2023) [51]	*n* = 8 (HFG) Age 72.37 ± 7.40 Male 87.5% H&Y stage 1.87 ± 0.35 *n* = 8 (LFG) Age 73.13 ± 2.85 Male 62.5% H&Y stage 1.44 ± 0.62	*n* = 8 Age 70.87 ± 7.77 Male 75% H&Y stage 2.19 ± 0.65	RCT, 12 wks HFG: strength training + WB-EMS 2x/wk LFG: AT + WB-EMS 2x/wk CG: UC strength training + WB-EMS: strength training (20 min) consisting of half squat, full squat, bent over, core rotation, and crunch, combined with WB-EMS (rectangular stimulation at 85 Hz, 350 µs, 4 s stimulation/4 s rest) AT + WB-EMS: aerobic training on rowing machine (20 min), combined with WB-EMS (rectangular stimulation 7 Hz 350 µs, with a continuous pulse duration) HRR 60–80%	HFG BDNF* 2131.5 pg/mL ± 628.0 à 1999.4 pg/mL ± 1074.9 (*p* > 0.05) LFG BDNF* 1426.5 pg/mL ± 1426.5 à 2042.2 pg/mL ± 567.5 (*p* > 0.05) CG BDNF* 1657.9 pg/mL ± 1035.9 à 862 pg/mL ± 760 (*p* < 0.05) Statistically significant change in BDNF between LFG and CG (*p* < 0.05) HFG 6MWT 280.5 m (49.3) à 267.5 m (41.25) (*p* > 0.05) LFG 6MWT 347 m (89.13) à 469 m (94.25) (*p* < 0.05) CG 6MWT 302 m (46.5) à 287.5 m (30.25) (*p* > 0.05) Statistically significant differences in 6MWT found in LFG compared to HFG and CG and HFG, and CG compared to LFG (*p* < 0.05) HFG Tinetti’s test 21.5 (4.5) à 21 (4) (*p* > 0.05) LFG Tinetti’s test 21.5 (3) à 28 (2) (*p* < 0.05) CG Tinetti’s test 22 (3.25) à 22.5 (2.5) (*p* > 0.05) Statistically significant differences in Tinetti’s test for balance found in LFG compared to HFG and CG HFG PFS-16 3.13 (0.45) à 2.8 (0.1) (*p* > 0.05) LFG PFS-16 3.23 (0.3) à 1.8 (0.25) (*p* < 0.05) CG PFS-16 3 (0.48) à 3.8 (0.47) (*p* > 0.05) Statistically significant differences in PFS-16 found in LFG compared to HFG and CG, and significant worsening compared with pre-test in CG	Exercise (specifically aerobic exercise) increased BDNF levels (no specific mechanism cited)
Frazzitta et al. (2014) [52]	*n* = 14 Age 67 ± 5 PD duration 8 ± 5 H&Y stage 1–1.5 UPDRS-III 16.4 ± 3.5	*n* = 10 Age 65 ± 4 PD duration 8 ± 2 H&Y stage 1–1.5 UPDRS-III 15.6 ± 1.5	RCT, 4 wks IG: IRT 5x/wk CG: UC IRT: stretching, warm-up, and ROM (60 min), balance, gait (30 min), treadmill plus (30 min), occupational therapy (60 min) ≤60% HRR Total: 180 min	IG BDNF 21,640 pg/mL ± 3400 à 24,770 pg/mL ± 6400 (*p* = 0.017) IG UPDRS-III 16.4 ± 3.5 à 8.8 ± 3.2 (*p* < 0.0001) IG UPDRS 25.43 ± 5.6 à 14.79 ± 6.0 (*p* < 0.0001) IG 6MWT 383 m ± 94 à 477m ± 79 (*p* = 0.0001) IG BBS 48.64 ± 6.1 à 54.00 ± 2.4 (*p* = 0.0016) CG BDNF 22870 pg/mL ± 4000 à 22580 pg/mL ± 4300 (*p* > 0.5) CG UPDRS-III 15.5 ± 1.4 à 15.8 ± 1.1 (*p* = 0.1934)No significant correlation between BDNF level change and changes in UPDRS-III (r = −0.13; *p* = 0.65), UPDRS (r = −0.18; *p* = 0.52), 6MWT (r = 0.05; *p* = 0.88), and BBS (r = −0.11; *p* = 0.69) ANOVA found significant inter-group differences in BDNF: F(3,66) = 5.63 (*p* = 0.0017) and UPDRS-III: F(3,66) = 66.5 (*p* < 0.001)	No mechanisms discussed
Freidle et al. (2022) [55]	*n* = 48 Age 71 ± 5.9 Male 62.5% PD duration 5.5 (7) H&Y stage 2.18 [2–3] UPDRS-III 31.2 ± 11.9	*n* = 47 Age 71.1 ± 6.3 Male 63.8% PD duration 3 (4) H&Y stage 2.28 [2–3] UPDRS-III 31.8 ± 10.3	RCT, 10 wks IG: HiBalance 2x/wk CG: HiCommunication 2x/wk HiBalance: group training focused on balance and cognitive and motor dual tasks (60 min) HiCommunication: training focused on improving speech and communication (60 min)	IG BDNF 38,010.8 pg/mL ± 7956.7 à 37169.4 pg/mL ± 5928.3 IG UPDRS 51.0 ± 18.8 à 48.2 ± 17.8 IG UPDRS-III 31.2 ± 11.9 à 29.1 ± 11.9 CG BDNF 37,805.3 pg/mL ± 8044.6 à 35,945.8 pg/mL ± 6208.5 CG UPDRS 50.4 ± 15.5 à 45.8 ± 16.8 CG UPDRS-III 31.8 ± 10.3 à 28.8 ± 10.7 No significant group X time interaction for BDNF outcome (*p* = 0.94) No significant group X time interaction for UPDRS outcome (*p* = 0.93) No significant group X time interaction for UPDRS-III outcome (*p* = 0.87)	No mechanisms discussed
O’Callaghan et al. (2020a) [53]	*n* = 13 Age 70.4 ± 7.217 Male 69.2% H&Y stage 2.07 [2–3]	*n* = 14 Age 64.6 ± 8.581 Male 57.1% H&Y stage 1.86 [1–2]	RCT, 12 wks IG: MICT 3x/wk CG: UC MICT: warm-up (10 min), aerobic exercise on treadmill (24 min), resistance training (12 min) 60–80% HR_max_ Total 45–60 min	IG BDNF* 1st session 1,433,133 pg/mL ± 605,390 à 1,626,033 pg/mL ± 861,609 IG BDNF* 12th session 1,457,900 pg/mL ± 606,219 à 1,481,967 pg/mL ± 1,441,211 Δ1st–12th (*p* = 0.650) CG BDNF* 1,386,000 pg/mL ± 1,192,620 à 989,233 pg/mL ± 1,024,793 (*p* = 0.140)	HIIT (as compared to aerobic exercise) induces hypoxia which triggers release of BDNF from cells
O’Callaghan et al. (2020b) [53]	*n* = 9 Age 68.8 ± 7.902 Male 55.6% H&Y stage 2.33 [2–3]	*n* = 8 Age 69.0 ± 6.633 Male 50% H&Y stage 2.25 [1–3]	RCT, 12 wks IG: HIIT 3x/wk CG: UC HIIT: warm-up (10 min), 4–6 × 4 min on Speedflex machine, 5 min cooldown ≥85% HR_max_ Total: 45–60 min	IG BDNF* 1st session 671,000 pg/mL ± 75,350 à 683,900 pg/mL ± 123,339 IG BDNF* 12th session 765,800 pg/mL ± 135,052 à 723,933 pg/mL ± 111,363 Δ1st–12th (*p* = 0.010) CG BDNF* 655,767 pg/mL ± 241,783 à 645,100 pg/mL ± 227,448 (*p* = 0.401)	HIIT (as compared to aerobic exercise) induces hypoxia which triggers release of BDNF from cells
Szymura et al. (2020) [54]	*n* = 16 (PDBT) Age 66.00 ± 2.59 Male 68,8% H&Y stage 2.44 [2–3] *n* = 16 (HBT) Age 67.25 ± 2.52 Male 62.5%	*n* = 13 (PDNT) Age 65.23 ± 7.40 Male 61.5% H&Y stage 2.31 [2–3] *n* = 16 (HNT) Age 65.69 ± 3.70 Male 62.5%	RCT, 12 wks PDBT: balance training 3x/wk PDNT: balance training 3x/wk HBT: UC HNT: UC Balance training: warm-up (5 min), balance training (50 min), cooldown (5 min) 60–70% HR_max_ Total 60 min	PDBT BDNF 21,190 pg/mL ± 8360 à 30370 pg/mL ± 6330 (*p* = 0.011) PDNT BDNF 30,080 pg/mL ± 8040 à 25780 pg/mL ± 11,720 (*p* > 0.05) HBT BDNF 20,210 pg/mL ± 13,330 à 34980 pg/mL ± 20,620 (*p* < 0.001) HNT BDNF 28,100 pg/mL ± 12,330 à 33,130 pg/mL ± 17,740 (*p* > 0.05)	Injured muscle (HIIT) recruit chemotactic factors (e.g., fractalkine) which induce release of BDNF from cells

Note: values are given in mean ± standard deviation (SD), mean [range], median (Inter Quartile Range), or percentage (%); BDNF is measured in serum unless stated otherwise. Abbreviations: wk(s) = week(s), min = minute, s = second, m = meter, pg/mL = picogram/milliliter, *n* = number, HR_max_ = maximum heart rate, HRR = heart rate reserve, IG = intervention group, CG = control group, UC = usual care, PD = Parkinson’s disease, BDNF = brain-derived neurotrophic factor, UPDRS = unified Parkinson’s disease rating scale, UPDRS-III = unified Parkinson’s disease rating scale motor examination, 6MWT = 6-min walk test, BBS = berg balance scale, H&Y = Hoehn and Yahr, ROM = range of motion, IRT = intensive rehabilitation treatment, MICT = moderate-intensity continuous training, HIIT = high-intensity interval training, PDBT = Parkinson’s disease with balance training, PDNT = Parkinson’s disease without training, HBT = healthy with balance training, HNT = healthy without training, HFG = high-frequency group, LFG = low-frequency group, WB-EMS = whole-body electromyostimulation, PFS-16 = Parkinson’s fatigue scale. * BDNF values are approximated from median (IQR) to mean ± SD [48]. Exercise was defined as follows: a subset of physical activity that is planned, structured, and repetitive and has as a final or an intermediate objective, which is the improvement or maintenance of physical fitness [57]. Column 1′s “Author (year)” displays the year of publication and the first author’s name of each included study. Column 2′s “Intervention” displays the participant characteristics of the intervention group of each included study. Column 3′s “Control” displays the participant characteristics of the control group of each included study. Column 4′s “Protocol/Exercise components” displays the (exercise) protocol for the intervention and control groups of each included study. Column 5′s “Results” displays the main results that are relevant to this review of each included study. Column 6′s “Proposed mechanism” displays the mechanism of BDNF change as a result of exercise that is discussed by authors of each included study.

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
