# Peer review of "Effects and Mechanisms of Exercise on Brain-Derived Neurotrophic Factor (BDNF) Levels and Clinical Outcomes in People with Parkinson’s Disease: A Systematic Review and Meta-Analysis"

_brainsci, 2024, doi:10.3390/brainsci14030194_

Round 1
Reviewer 1 Report
Comments and Suggestions for Authors
The manuscript presents a systematic review and meta-analysis focusing on the effects of exercise therapy on brain-derived neurotrophic factor (BDNF) levels and clinical outcomes in individuals with Parkinson's disease (PD). The authors conducted a comprehensive literature search and included randomized clinical trials comparing exercise interventions to usual care, sham interventions, or no intervention. The results of their meta-analysis indicate a significant increase in BDNF levels among participants who underwent exercise therapy, along with improvements in various clinical outcomes such as motor symptoms, walking ability, and balance. The authors discuss the potential mechanisms underlying the observed effects and highlight the neuroplastic effects of exercise on the Parkinsonian brain, partly mediated by BDNF.
The methodological quality of the included trials is not consistently addressed or discussed.
The manuscript lacks a clear description of the specific exercise interventions employed in the included trials.
Further discussion on potential limitations and sources of heterogeneity in the meta-analysis would enhance the manuscript.
The manuscript would benefit from a more in-depth exploration of the clinical implications and practical applications of the findings.
The study makes a valuable contribution to the field by providing evidence for the positive effects of exercise therapy on BDNF levels and clinical outcomes in individuals with Parkinson's disease. However, the manuscript requires substantial revisions to address the following points:
Methodological Quality Assessment: Provide a thorough assessment of the methodological quality of the included trials. Utilize established quality assessment tools, such as the Cochrane Risk of Bias tool, to evaluate the risk of bias in individual studies. Discuss the strengths and weaknesses of each study in terms of randomization, blinding, allocation concealment, and other relevant aspects.
Detailed Intervention Description: Clearly describe the specific exercise interventions used in the included trials. Include information on exercise duration, intensity, frequency, and any modifications or adaptations made for participants with Parkinson's disease. This will help readers understand the nature of the interventions and their potential impact on BDNF levels and clinical outcomes.
Discussion of Heterogeneity: Discuss the potential sources of heterogeneity among the included studies, such as variations in participant characteristics, study design, intervention protocols, and outcome measures. Use subgroup analyses or sensitivity analyses, if appropriate, to explore the impact of these factors on the overall results.
Clinical Implications: Provide a more comprehensive discussion of the clinical implications of the findings. Explore how the observed increase in BDNF levels and improvements in clinical outcomes may translate into meaningful benefits for individuals with Parkinson's disease. Discuss the potential integration of exercise therapy into clinical practice and the implications for healthcare professionals and patients.
Limitations: Discuss the limitations of the study, including any potential biases or limitations in the included trials. Address the generalizability of the findings and any potential confounding factors that may have influenced the results.
Future Research Directions: Provide recommendations for future research based on the findings of this systematic review and meta-analysis. Identify gaps in the current literature and suggest areas for further investigation to strengthen the evidence base.
Abstract Revision: Revise the abstract to provide a concise summary of the key findings, methodology, and implications of the study. Ensure that the abstract accurately reflects the revised content of the manuscript.
Formatting and Language: Ensure consistency in the formatting of references throughout the manuscript. Proofread the manuscript for any language errors or typos.
Reviewer 2 Report
Comments and Suggestions for Authors
1. What is the take-home message of this manuscript?
2. Why did the author evaluate only one database? The Scopus, EMBAS, and Scholar needs.
3. Describe the limitations and suggestions for future study.
4. This study needs to evaluate the risk of bias and quality assessment of included evidence.
5. Table 3 is unclear, and the reader is confused.
6. The distribution of evidence is insufficient.
Reviewer 3 Report
Comments and Suggestions for Authors
The purpose of this systematic review and meta-analysis was to give an updated overview of the mechanisms and effects of exercise therapy on BDNF levels in PD, in order to bring these insights into the clinical context of physical medicine and rehabilitation for people living with PD. The authors felt this was needed and timely as previous meta-analyses had conflicting results on this topic.
Overall, the was conducted carefully and all aspects of the methodology were solid. The writing was good, easy to understand, and had very few grammatical, formatting, or typographical errors. The study adds to the literature on the topics of BDNF and exercise and PD. The study should be of interest to readers of the journal and researchers in several adjacent fields.
I only have minor comments and concerns.
- Is 5 studies enough for a meta-analysis, the authors should give more reasoning and rationale about why this is not and issue and the meta-analysis is needed, other than the problems with past meta-analyses. Especially since most included studies also had small sample sizes.
- Bibliography has some errors. The titles of some articles have the first letter capitalized for all words, whereas others do not. The names of journals have inconsistencies in capitalization etc also.
- Line 27 was AND capitalized on purpose?
- Lines 261, 274 and some others in the results and discussion sections are paragraphs of only 1 sentence, shouldn’t these be combined with previous paragraphs so as not to have 1 sentence paragraphs.
Comments on the Quality of English Language
minor proofreading, bibliography errors
Round 2
Reviewer 2 Report
Comments and Suggestions for Authors
The author responds to my comments.
Author Response
Dear reviewer,
The comments you have provided have been adressed in the attached word file in a point-to-point manner.
